# LongRAG: Enhancing Retrieval-Augmented Generation with Long-context LLMs

## Abstract

In traditional RAG framework, the basic retrieval units are normally short. The common retrievers like DPR normally work with 100-word Wikipedia paragraphs. Such a design forces the retriever to search over a large corpus to find the "needle" unit. In contrast, the readers only need to generate answers from the short retrieved units. The imbalanced "heavy" retriever and "light" reader design can lead to sub-optimal performance. The loss of contextual information in the short, chunked units may increase the likelihood of introducing hard negatives during the retrieval stage. Additionally, the reader might not fully leverage the capabilities of recent advancements in LLMs. In order to alleviate the imbalance, we propose a new framework LongRAG, consisting of a **"long retriever"** and a **"long reader"**. In the two Wikipedia-based datasets, NQ and HotpotQA, where the average document size is less than 1K tokens, LongRAG processes the entire Wikipedia corpus into 4K-token units by grouping related documents, making these units 30 times longer than before. By increasing the unit size, we significantly reduce the total number of units from 22M to 600K. This greatly reduces the burden on the retriever, resulting in strong retrieval performance with only a few (less than 8) top units. Compared to traditional RAG, which may require hundreds of short units to achieve similar retrieval performance, our approach minimizes the likelihood of retrieving hard negatives while maintaining semantic integrity of each unit. Then we feed these retrieved units ($\approx$ 30K tokens) to an existing long-context LLM to perform zero-shot answer generation. Without requiring any training, LongRAG achieves an EM of 62.7% on NQ and 64.3% on HotpotQA, which are on par with the (fully-trained) SoTA model. Furthermore, we test on two non-Wikipedia-based datasets, Qasper and MultiFieldQA-en, where the average document length is already above 4K tokens. LongRAG processes each individual document as a single (long) unit rather than chunking them into smaller units. By doing so, we achieve an F1 score of 25.9% on Qasper (previously 22.5%) and 57.5% on MultiFieldQA-en (previously 51.2%). Our study offers insights into the future roadmap for combining RAG with long-context LLMs.

## 1 Introduction

Retrieval-Augmented Generation (RAG) methods have long been employed to enhance large language models (LLMs) (Mialon et al., 2023). Knowledge in the form of natural language can be entirely offloaded from the parametric knowledge of LLMs by leveraging a standalone retrieval component from an external corpus. The existing RAG framework tends to use short retrieval units, such as 100-word passages in popular open-domain question-answering tasks (Chen et al., 2017; Lewis et al., 2020; Karpukhin et al., 2020). The retriever is tasked with finding the "needle" (i.e. the precise tiny retrieval unit) from the "haystack" (i.e. the massive corpus with up to tens of millions of information units). Subsequently, the retrieved units are passed to the reader to generate the final response. On the contrary, the reader only needs to extract answers from these retrievals, which is a fairly easy task. This kind of imbalanced design, with a "heavy" retriever and a "light" reader, puts too much pressure on the retriever. Therefore, existing RAG models (Izacard & Grave, 2020b) have to recall huge amounts of units, such as the top-100/200, combined with additional re-ranker to achieve the best performance. Moreover, short retrieval units can lead to semantic incompleteness due

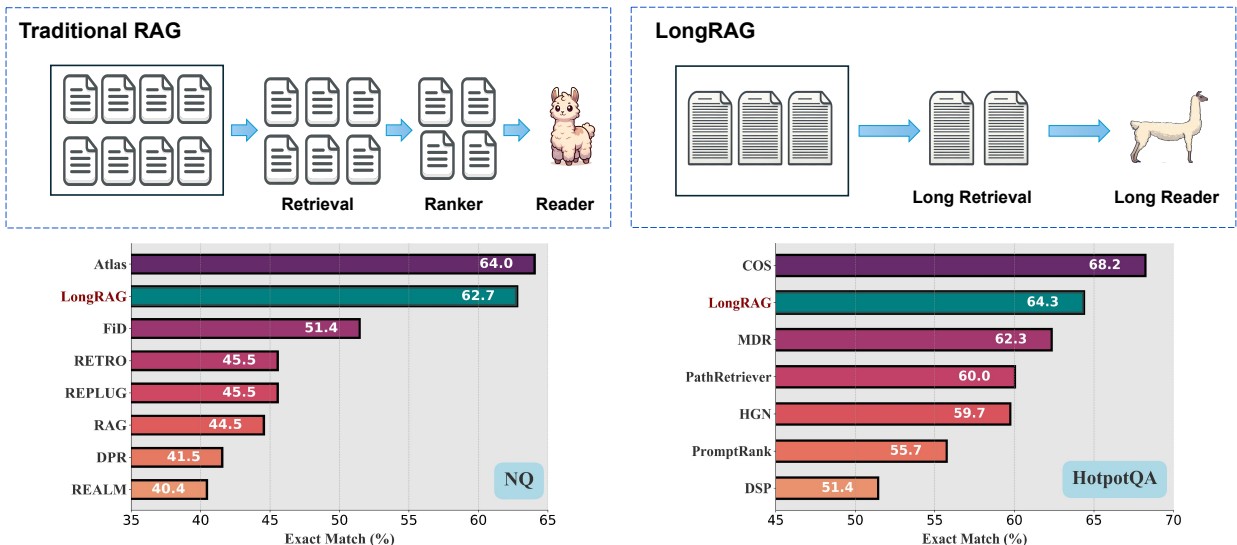

Figure 1: Traditional RAG vs. LongRAG. (Up) Traditional RAG operates on short retrieval units, where the retriever needs to scan over a massive amount of units to find the relevant piece. In contrast, LongRAG operates on long retrieval units (30x longer). The retriever of LongRAG has a significantly reduced workload, achieving strong retrieval quality by leveraging only a few top units without the need for additional ranking mechanisms or other complex components. LongRAG could fully exploit the ability of long-context language models to achieve strong performance. (Down) QA performance compared with other methods on the NQ dataset and the HotpotQA dataset.

to document truncation. This can result in the loss of contextual information, which may ultimately harm overall performance. This design choice was made in an era when the reader models were heavily restricted by their ability to handle long and contexts. With the recent advances in long-context language models, the reader can potentially handle up to 128K or even millions of tokens as input (Reid et al., 2024; Achiam et al., 2023). In this paper, we propose to revisit this design choice for open-domain question answering and propose the LongRAG framework as a solution to balance the workload between the retriever and the reader, as illustrated in Figure 1. There are three important designs in our novel framework:

**Long Retrieval Unit**: By using entire documents or grouping multiple related documents, we can construct long retrieval units with more than 4K tokens. This design could also significantly reduce the corpus size (number of retrieval units in the corpus). This makes the retriever's task much easier by providing more complete information, allowing the retriever's architecture to be simplified without the need for additional re-rankers or iterative retrieval.

**Long Retriever**: The long retriever will identify coarse relevant information for the given query by searching through all the long retrieval units in the corpus. Only a few top retrieval units (1 to 8 retrieval units in the four datasets we tested on), without re-ranking, are used for the next step. Compared to retrieving hundreds of short units, the long retriever only needs to retrieve a few candidates, which significantly reduces the likelihood to encounter hard negatives (it will confuse the reader).

**Long Reader**: The long reader will further extract answers from the concatenation of retrievals, which is normally around 30K tokens. We simply prompt an existing long-context LM (like Gemini or GPT4) with the question to produce the answers in a zero-shot fashion.

These three novel designs significantly boost the overall performance of RAG on open-domain question-answering tasks like NQ (Kwiatkowski et al., 2019), HotpotQA (Yang et al., 2018), Qasper (Dasigi et al., 2021) and MultiFieldQA-en (Bai et al., 2023).

In our experiments, we adopt off-the-shelf retrievers like BGE (Xiao et al., 2023) and readers like Gemini-1.5-Pro (Reid et al., 2024) or GPT-4o (OpenAI, 2024) without any further tuning. To demonstrate the

generalizability of our proposed framework, we tested it on four datasets from different scenarios. First, we evaluate it on NQ and HotpotQA, which are Wikipedia-based dataset. The corpus of both datasets are composed of relatively short (averaging less than 1K tokens) but vast Wikipedia documents. By forming longer retrieval units through the grouping of multiple related documents, we reduce the NQ corpus size from 22M to 600K units, which improves the answer recall@1 from 52% (DPR) to 71%. Similarly, we reduce the HotpotQA corpus size from 5M to 500K, which improves the recall@2 from 47% (DPR) to 72%. By exploiting the long-context understanding ability of GPT-4o, LongRAG can achieve an EM of 62.7% on NQ and 64.3% on HotpotQA. These results could be comparable to the strongest fully trained RAG models like Atlas (Izacard et al., 2022) and MDR (Xiong et al., 2020b). Furthermore, we test on two non-Wikipedia-based datasets, Qasper and MultiFieldQA-en, where the corpus consists of relatively long documents averaging more than 4K tokens. LongRAG processes each entire document as a single unit rather than chunking them into smaller units. By doing so, we achieve an F1 score of 25.9% on Qasper (previously 22.5%) and 57.5% on MultiFieldQA-en (previously 51.2%).

We perform ablation studies in subsection 3.5 to prove why longer retrieval units are necessary. Given a budget of 40K recall tokens, with "short retriever units", we can increase the number of recalled units to reach a marvelously high recall score (91% for recall@200). However, the end performance dips significantly due to the huge amount of "hard negatives", which confuses the reader. With "long retriever units", we observe an entirely different trend. As we recall more units (from 1 to 8 units), both the recall and end performance will increase or plateau. The impact of "hard negative" is much less severe in LongRAG. It shows that LongRAG can better exploit the advances in the long-context LLMs (reader). As the long-context methods evolve, the performance of LongRAG will continue to improve. Therefore, we believe the modern RAG systems should re-consider the granularity of their retrieval units to exploit the advantages of the current long-context LLMs.

## 2 LongRAG

Our proposed LongRAG framework is comprised of two components: the **Long Retriever** and the **Long Reader**. Compared to traditional RAG, which operates on a large number of short retrieval units, LongRAG operates on long retrieval units, with only a few (typically fewer than 10) being fed into the reader. An illustrative example is shown in Figure 2.

### 2.1 Long Retriever

The traditional RAG framework employs smaller retrieval units and prioritizes retrieving the exact fine-grained short context containing the answer. In contrast, our proposed LongRAG framework places greater emphasis on recall, aiming to retrieve relevant context with much coarse granularity. This design choice shifts more burden from the retriever to the reader to extract the exact answers from the relevant context.

We denote our corpus for retrieval as $\mathcal{C} = \{d_1, d_2, \ldots, d_D\}$, which is a collection of $D$ documents. Formally speaking, the long context retriever is a function: $\mathcal{F} : (q, \mathcal{C}) \rightarrow \mathcal{C}_{\mathcal{F}}$ that takes as input a question $q$ and a corpus $\mathcal{C}$ and returns a filtered set of texts $\mathcal{C}_{\mathcal{F}} \subset \mathcal{C}$. In traditional RAG, $\mathcal{C}_{\mathcal{F}}$ is usually small which contains about hundred of tokens, which should contain exact information related to the question $q$. In our framework, $\mathcal{C}_{\mathcal{F}}$ is usually more than 4K tokens, which contains relavant but not exact information related to the question $q$. The long retriever function $\mathcal{F} : (q, \mathcal{C})$ is then divided into three steps:

**Formulate long retrieval units** A function is applied to the corpus to form $M$ retrieval units: $\mathcal{G}(\mathcal{C}) = \{g_1, g_2, \ldots, g_M\}$. In traditional RAG, the retrieval unit $g$ is typically a short span of passage which is split from the documents $d$, containing hundreds of tokens. In our framework, $g$ could be as long as the whole document or even a group of documents, resulting in much longer retrieval units. If the original document is already long (e.g., longer than 4K tokens), we treat the entire document as a single unit. If the original document is relatively short (e.g., shorter than 1K tokens), we group related documents together to form a single unit. We provide an example of a grouping algorithm in Appendix A.3.

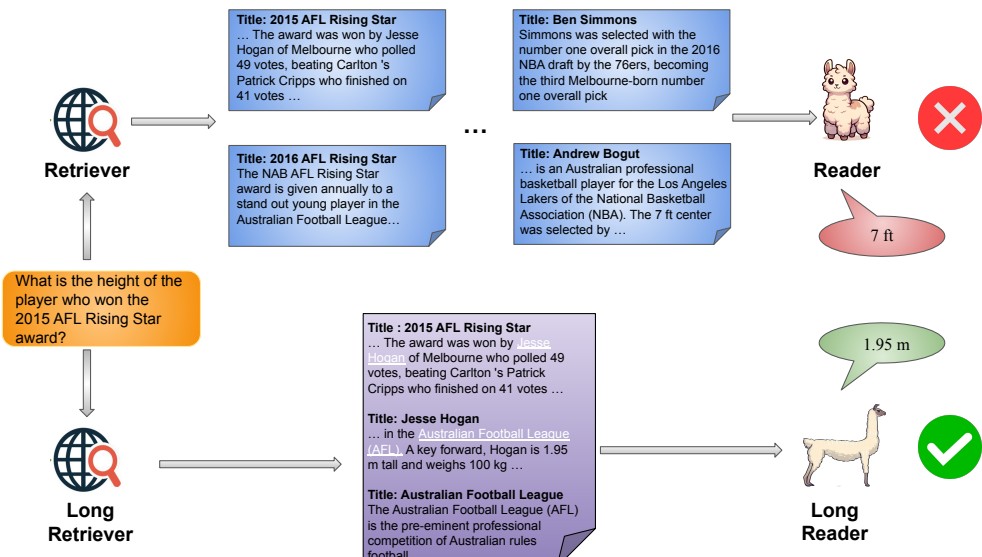

Figure 2: LongRAG example. We form the long retrieval unit which is at least 4K tokens by using the entire document or grouping related documents, depending on the orinal docuemnt size. In this example, multiple Wikipedia documents are grouped through hyperlinks. This approach enables even multi-hop question-answering cases from HotpotQA to be addressed using only a few retrieval units, which are then fed into a long reader. Compared to traditional RAG, which retrieves hundreds of short units, our proposed LongRAG reduces the likelihood of retrieving hard negatives during the retrieval stage and more effectively leverages recent advances in long-context LLMs.

By having a longer retrieval unit, there are two key advantages: First, only very few (e.g., 4 to 8 retrieval units) are fed into the reader, which greatly reduces the likelihood of encountering hard negatives compared to traditional RAG, which may require hundreds of short units in its reader. Second, by retaining the entire document or even related documents within a single retrieval unit, the contextual information is preserved.

**Similarity search** We utilize an encoder, denoted as $E_Q(\cdot)$, to map the input question to a $d$-dimensional vector. Additionally, we employ another encoder, $E_C(\cdot)$, to map the retrieval unit to a $d$-dimensional vector. We define the similarity between the question and the retrieval unit using the dot product of their vectors:

$$sim(q, g) = E_Q(q)^T E_C(g)$$

In LongRAG settings, $E_C(g)$ is challenging given the length of $g$, so we resort to an approximation as below.

$$sim(q, g) = E_Q(q)^T E_C(g) \approx \max_{g' \subseteq g}(E_Q(q)^T E_C(g'))$$

We approximate it by maximizing the scores of all chunks $g'$ within the retrieval unit $g$, akin to the MaxP design in (Dai & Callan, 2019). We consider different levels of granularity of chunk $g'$, including 512 tokens, 4K tokens, and encoding the entire $g$ completely. The empirical study about this settings is in Table 4. With this similarity score setup, we will retrieve the top $k$ retrieval units closest to the given query. For efficient retrieval, we precompute the embedding of each retrieval unit $g'$ and predict the exact inner product search index in FAISS (Johnson et al., 2019).

**Aggregate retrieval result** We will concatenate the top $k$ retrieval units into the long context as the retrieval result, denoted by $\mathcal{C}_{\mathcal{F}} = \text{Concat}(g^1, g^2, \ldots, g^k)$. Depending on the selection of retrieval units, a larger retrieval unit size will result in a smaller value of $k$ being used. For instance, in NQ dataset, if the retrieval unit is a passage, $k$ is approximately above 100; if it's a document, $k$ is around 10; and for grouped documents as retrieval units, we typically set $k$ to 4 to 8.

## 2.2 Long Reader

The long reader operates straightforwardly. We feed the related instruction $i$, the question $q$, and the long retrieval result $C_{\mathcal{F}}$ into an LLM, enabling it to reason over the long context and generate the final output. It's important that the LLM used in the long reader can handle long contexts and does not exhibit excessive position bias. We select Gemini-1.5-Pro (Reid et al., 2024) and GPT-4o (OpenAI, 2024) as our long reader given their strong ability to handle long context input. We utilize different approaches for short and long contexts. For short contexts, typically containing fewer than 1K tokens, we instruct the reader to directly generate the answer from the provided context retrieved from the corpus. For long contexts, typically longer than 4K tokens, we empirically find that using a similar prompt as for short contexts, where the model extracts the final answer directly from the long context, often leads to decreased performance. Instead, the most effective approach is to utilize the LLM as a chat model. Initially, it outputs a long answer, typically spanning a few words to a few sentences. Subsequently, we prompt it to generate a short answer by further extracting it from the long answer. The prompt is provided in the Appendix A.1.

## 3 Experiments

In this section, we will first provide detailed descriptions of the four datasets we use, followed by a demonstration of the retriever's performance. Next, we will present the final question-answering performance. Finally, we conduct detailed ablation studies to explain why operating on long retrieval units benefits performance.

### 3.1 Dataset

Our proposed methods were tested on four question-answering datasets. The basic statistics are shown in Table 1. Additionally, we have provided some examples in Appendix A.4.

| Dataset | Corpus source | Avg. Doc. Length | # of Documents | # of Text cases | Metric |
|---|---|---|---|---|---|
| NQ | Wikipedia | 800 | 3M | 3,610 | EM |
| HotpotQA | Wikipedia | 130 | 5.2M | 7,405 | EM |
| Qasper | Science | 4.7K | 416 | 371 | F1 |
| MultiFieldQA-en | Multi-field | 6.9K | 150 | 150 | F1 |

Table 1: An overview of the four datasets used in our experiments is provided. "Corpus source" refers to the origin of the retrieval corpus. We selected NQ and HotpotQA from Wikipedia, Qasper from scientific documents, and MultifieldQA-en from multi-field documents. The two Wikipedia-based datasets utilize a massive retrieval corpus containing millions of short documents. In contrast, the other two datasets employ a smaller corpus consisting of hundreds of long documents.

**Natural Question** (Kwiatkowski et al., 2019) is designed for end-to-end question answering. The questions are mined from real Google search queries and the answers were spans in Wikipedia articles identified by annotators. This dataset contains 3,610 questions. For NQ, we use the Wikipedia dumps from December 20, 2018, which contain approximately 3 million documents and 22 million passages.

**HotpotQA** (Yang et al., 2018) consists of two-hop questions over diverse topics. We focus on the fullwiki setting in which two Wikipedia passages are required to answer the questions. Since the gold passages for the test set are not available, we follow prior work (Xiong et al., 2020b) and evaluate on the development set, which has 7,405 questions. There are two main question types in HotpotQA: (1) comparison questions usually require contrasting two entities and (2) bridge questions can be answered by following a connecting entity that links one document to another. For HotpotQA, we use the abstract paragraphs from the October 1, 2017 dump, which contain around 5 million documents.

**Qasper** (Dasigi et al., 2021) is an information-seeking question answering dataset over academic research papers. Each question is written as a followup to the title and abstract of a particular paper, and the answer, if present, is identified in the rest of the paper. The original Qasper dataset is a single-document QA dataset.

We refactor it into a RAG task, where the system first retrieves the necessary document and then answers the given question, following a design similar to LoCoV1 (Saad-Falcon et al., 2024).

**MultifieldQA-en** (Bai et al., 2023) is a question-answering dataset based on long documents from diverse sources, including legal documents, government reports, encyclopedias, and academic papers. The original MultifieldQA-en is a single-document QA dataset. We refactor the dataset into a RAG task, where the system first retrieves the necessary document and then answers the given question, following a design similar to LoCoV1 (Saad-Falcon et al., 2024).

## 3.2 Retrieval Performance

In this section, we present the retrieval performance on two extractive QA datasets, NQ and HotpotQA, to demonstrate that comparable retrieval performance can be achieved using only a few long retrieval units (such as 4 to 8). This approach contrasts with the use of hundreds of short retrieval units, which may lead to information loss and the introduction of hard negatives that can confuse the reader and prevent the full utilization of long-context LLMs. For the other two datasets, it's not straightforward to compare retrieval performance at different granularities since they are not extractive QA tasks. Therefore, we will directly discuss the final QA results in the next section.

**Metrics**   Retrieval performance is measured using Answer Recall (AR) and Recall (R). For NQ, we use only answer recall, while for HotpotQA, we use both metrics. Answer Recall is the recall of the answer string in all the retrieved documents that we plan to use in the reader. For example, if the retrieval unit is at the "passage" level and the number of retrieval units is 100, answer recall measures whether the answer string is present in these 100 passages. For HotpotQA, we compute AR only for questions with span answers, specifically the "bridge" type questions, while ignoring yes/no and comparison questions, following previous work (Khalifa et al., 2022). Recall used for HotpotQA measures whether the two gold documents are present in all the retrieved results. For example, if the retrieval unit is at the "document" level and the number of retrieval units is 10, recall measures whether both gold documents are present among the 10 retrieval.

**Experiment Setup**   We leverage open-sourced dense retrieval toolkit, Tevatron (Gao et al., 2022), for all our retrieval experiments. The base embedding model we used is bge-large-en-v1.5, a general-purpose embeddings model that isn't specifically trained on our test data.

| Retrieval Unit | Corpus Size | Num of Retrieval Units | Average Num of Tokens | | Answer Recall (AR) |
|---|---|---|---|---|---|
| | | | Corpus | Test Set | |
| Passage | 22M | 1 | 120 | 130 | 52.24 |
| | | 100 | 12K | 14K | 89.92 |
| | | 200 | 24K | 28K | 91.30 |
| Document | 3M | 1 | 820 | 4K | 69.45 |
| | | 5 | 4K | 18K | 85.37 |
| | | 10 | 8K | 34K | 88.12 |
| Grouped Documents | 600K | 1 | 4K | 6K | 71.69 |
| | | 4 | 16K | 25K | 86.30 |
| | | 8 | 32K | 50K | 88.53 |

Table 2: The table illustrates the retrieval performance on NQ. Employing a long-context retriever (with an average number of tokens for each retrieval unit up to 6K) compresses the corpus size by up to 30 times (from 22M to 600K), enhancing top-1 answer recall by approximately 20 points (from 52.24 to 71.69). Furthermore, long-context retriever requires significantly fewer retrieval units (10x fewer) to achieve comparable results. Therefore, integrating long-context retrieval significantly alleviates the burden of retriever.

Table 2 and Table 3 have shown the retrieval results on NQ and HotpotQA. In the NQ dataset, we utilize three different retrieval units, ranging from shorter to longer: passage, document, and grouped documents. In the table, we have mentioned two kinds of average number of tokens in each retrieval unit: one for the entire corpus and one for each test set. The retrieval units for each test case can sometimes be much longer than the average size across the whole corpus, as the corpus might include some Wikipedia pages with very

| Retrieval Unit | Corpus Size | Num of Retrieval Units | Average Num of Tokens | | Recall (R) | Answer Recall (AR) |
|---|---|---|---|---|---|---|
| | | | Corpus | Test Set | | |
| Document | 5.2M | 2 | 130 | 200 | 30.01 | 47.75 |
| | | 100 | 6.5K | 10K | 74.84 | 84.67 |
| | | 200 | 13K | 20K | 79.68 | 88.34 |
| Grouped Documents | 500K | 2 | 1K | 8K | 56.30 | 72.49 |
| | | 8 | 4K | 29K | 74.71 | 84.40 |

Table 3: The table illustrates the retrieval performance on HotpotQA. Similar to the findings on NQ, a long-context retrieval could significantly alleviate the burden on the retriever component.

few words, while the test cases may focus more on longer documents. Generally, our long-context retriever (at the document level and grouped document level) uses retrieval units containing an average of 6K tokens. By using longer retrieval units, there are several advantages: 1) It will significantly alleviate the burden on the retriever by compressing the corpus size by approximately 30 times, from 22M to 600K. The top-1 answer recall improves by about 20 points, from 52.24 to 71.69. We could use significantly fewer retrieval units to achieve comparable retrieval performance. For instance, 8 retrieval units at the grouped document level can achieve similar recall as 100 retrieval units at the passage level. 2) It could provide more comprehensive information to the reader. In the original passage-level RAG setup, information might be incomplete due to the chunking operation. In the HotpotQA dataset, we observe similar results. One notable difference is that in HotpotQA, the retrieval units are only at the document level and grouped document level, as HotpotQA uses only abstract paragraphs from each Wikipedia page.

| Model | Granularity | AR@1 |
|---|---|---|
| BGE-Large | 512-tokens chunk | 71.7% |
| E5-Mistral-7B | 4000-tokens chunk | 54.2% |
| E5-Mistral-7B | entire grouped retrieval unit | 23.4% |

Table 4: Different methods to encode the long retrieval unit in the long retriever. Using a general embedding model and approximating by maximizing the similarity scores between the query and all chunks within the retrieval unit is better than using the existing long embedding model to encode the entire context.

**Encode the long retrieval unit** As discussed in Section 2.2, it's very challenging to employ an encoder, $E_C(\cdot)$, to map the retrieval unit $g$ to a $d$-dimensional vector when $g$ is very long. Therefore, we use an approximation in our proposed system. Table 4 demonstrates that our approximation, $sim(q, g) = E_Q(q)^T E_C(g) \approx \max_{g' \subseteq g}(E_Q(q)^T E_C(g'))$, is much more effective than encoding the entire long context directly. We compare three methods: 1) Using the general embedding model "bge-large-en-v1.5" (Xiao et al., 2023), with $g'$ selected as text of 512-token size. 2) Using long embedding model "E5-Mistral-7B" (Zhu et al., 2024a), with $g'$ selected as the whole document, which has an average size of 4K tokens. 3) Using long embeddings model "E5-Mistral-7B", with no approximation, we encode the entire $g$, which is composed of multiple documents, directly. The average size of $g$ is 6K tokens. We can notice from the table that our approximation by taking the maximum score between the query and each text piece from the long context produces much better results than encoding them directly using the long embedding model. We believe that future advancements in long embedding models, which focus on encoding long contexts or multiple documents, will further enhance our framework and reduce memory consumption.

## 3.3 Full QA Performance on Wikipedia-based Datasets

We leverage Gemini-1.5-Pro and GPT-4o as the reader in our LongRAG framework. The prompt we use for our experiments are in Table 7. For Wiki-based datasets, such as NQ and HotpotQA, which generate short answers typically less than 5 tokens, we use EM (Exact Match rate) as the evaluation metric. We also refine the standard exact match rate definition to more fairly evaluate LongRAG's performance. More details can be found in Section A.2.

| NQ | EM | | HotpotQA | EM |
|---|---|---|---|---|
| *Closed-Book* | | | *Closed-Book* | |
| GPT-4-Turbo (Achiam et al., 2023) | 41.2 | | Claude-3-Opus (Anthropic, 2024) | 32.8 |
| Gemini-1.5-Pro (Reid et al., 2024) | 47.8 | | Gemini-1.5-Pro (Reid et al., 2024) | 33.9 |
| Claude-3-Opus (Anthropic, 2024) | 49.2 | | GPT-4-Turbo (Achiam et al., 2023) | 42.4 |
| *Fully-supervised RAG* | | | *Fully-supervised RAG* | |
| REALM (Guu et al., 2020) | 40.4 | | DrKIT (Dhingra et al., 2020) | 42.1 |
| DPR (Karpukhin et al., 2020) | 41.5 | | Transformer-XH (Zhao et al., 2019) | 51.6 |
| RAG (Lewis et al., 2020) | 44.5 | | QAMAT+ (Chen et al., 2023b) | 57.6 |
| RETRO (Borgeaud et al., 2022) | 45.5 | | HGN (Fang et al., 2019) | 59.7 |
| RePAQ (Lewis et al., 2021) | 47.8 | | PathRetriever (Asai et al., 2019) | 60.0 |
| FID (Izacard & Grave, 2020b) | 51.4 | | HopRetrieve (Li et al., 2021) | 62.1 |
| EMDR$^2$ (Singh et al., 2021) | 52.5 | | MDR (Xiong et al., 2020b) | 62.3 |
| FID-KD (Izacard & Grave, 2021) | 54.7 | | HopRetrieve-plus (Li et al., 2021) | 66.5 |
| R2-D2 (Fajcik et al., 2021) | 55.9 | | AISO (Zhu et al., 2021) | 68.1 |
| Atlas (Izacard et al., 2022) | 64.0 | | COS (Ma et al., 2023) | 68.2 |
| *No Fine-tuning RAG* | | | *No Fine-tuning RAG* | |
| REPLUG (Shi et al., 2023) | 44.7 | | DSP (Khattab et al., 2022) | 51.4 |
| REPLUG + LSR (Shi et al., 2023) | 45.5 | | PromptRank (Khalifa et al., 2023) | 55.7 |
| LongRAG (Gemini-1.5-Pro; Recall 4 units) | 58.6 | | LongRAG (Gemini-1.5-Pro; Recall 8 units) | 57.5 |
| LongRAG (GPT-4o; Recall 4 units) | **62.7** | | LongRAG (GPT-4o; Recall 8 units) | **64.3** |

Table 5: The tables show the QA results on the NQ test dataset (left) and Hotpot-QA dev set (right). We compare the results with three groups of baselines: closed-book, which involves directly prompting state-of-the-art LLMs with 16-shot in-context examples; fully-supervised RAG, where the RAG framework is used and the model is fully supervised and trained on the training data; and No Fine-tuning RAG, which employs the RAG framework without any tuning.

For NQ and HotpotQA, we compare our model with several groups of strong previous models as baselines. The first group is "**Closed-Book**": These baselines mean that no retrieval component is used; instead, state-of-the-art LLMs are employed to directly obtain the final result. We evaluate our results on Gemini-1.5-pro (Reid et al., 2024), Claude-3-Opus (Anthropic, 2024) and GPT-4-Turbo (Achiam et al., 2023). All models are evaluated on 16-shot in-context learning with direct prompting; The second group is "**Fully-supervised RAG**", and these baselines involve full-supervised fine-tuning on the training dataset. The third group is "**No Fine-tuning RAG**", and these baselines doesn't involve any supervised fine-tuning. The QA results on NQ and HotpotQA are presented in Table 5. On the NQ dataset, LongRAG achieves a 62.7 exact match rate, which is on par of the strongest fine-tuned RAG model like Atlas. On the HotpotQA dataset, LongRAG achieves a 64.3 exact match rate, which is also close to the SoTA fully-supervised RAG frameworks.

### 3.4 Full QA Performance on non-Wikipedia-based Datasets

For datasets that generate long answers, such as Qasper and MultifieldQA-en, we use the token-level F1 score (F1) as the evaluation metric. For Qasper and MultifieldQA-en, since we repurpose the datasets from single-document QA to a RAG task, we do not directly compare the results with previous models. Instead, we compare the performance of traditional RAG, which operates on 200-token passages, with our LongRAG, which operates on entire documents ranging from 4K to 6K tokens. The results are shown in Table 6. We observe that using long retrieval units at the whole document level performs better than using hundreds of short chunked retrieval units. On the Qasper dataset, gathering 100 short retrieval units of 200 tokens each into the reader achieves a 22.6% F1 score, while using a single long retrieval unit of 5K tokens achieves a 26.3% F1 score. Similarly, on the MultifieldQA-en dataset, gathering 100 short retrieval units of 200 tokens each into the reader results in a 51.3% F1 score, whereas using five long retrieval units of 7K tokens each results in a 57.5% F1 score.

| Retrieval Unit | Num of Retrieval Units | Qasper | MutilfieldQA-en |
|---|---|---|---|
| Passage | 1 | 15.5 | 38.9 |
| | 10 | 20.6 | 47.3 |
| | 100 | 22.6 | 51.3 |
| | 200 | 21.9 | 50.9 |
| Document | 1 | **26.3** | 49.4 |
| | 2 | 25.9 | 50.2 |
| | 5 | 23.9 | **57.5** |
| | 10 | 21.6 | 56.8 |

Table 6: This table presents the QA results on two non-Wiki datasets: Qasper and MultifieldQA-en. The results are evaluated based on token-level F1. Both datasets contain long documents, averaging at least 4K tokens. The results demonstrate that our LongRAG, which operates on long retrieval units, achieves better performance compared to traditional RAG, which operates on short retrieval units.

## 3.5 Ablation Studies

We perform several in-depth ablation to understand what are the important factors in our LongRAG system including "unit size" and "reader variant".

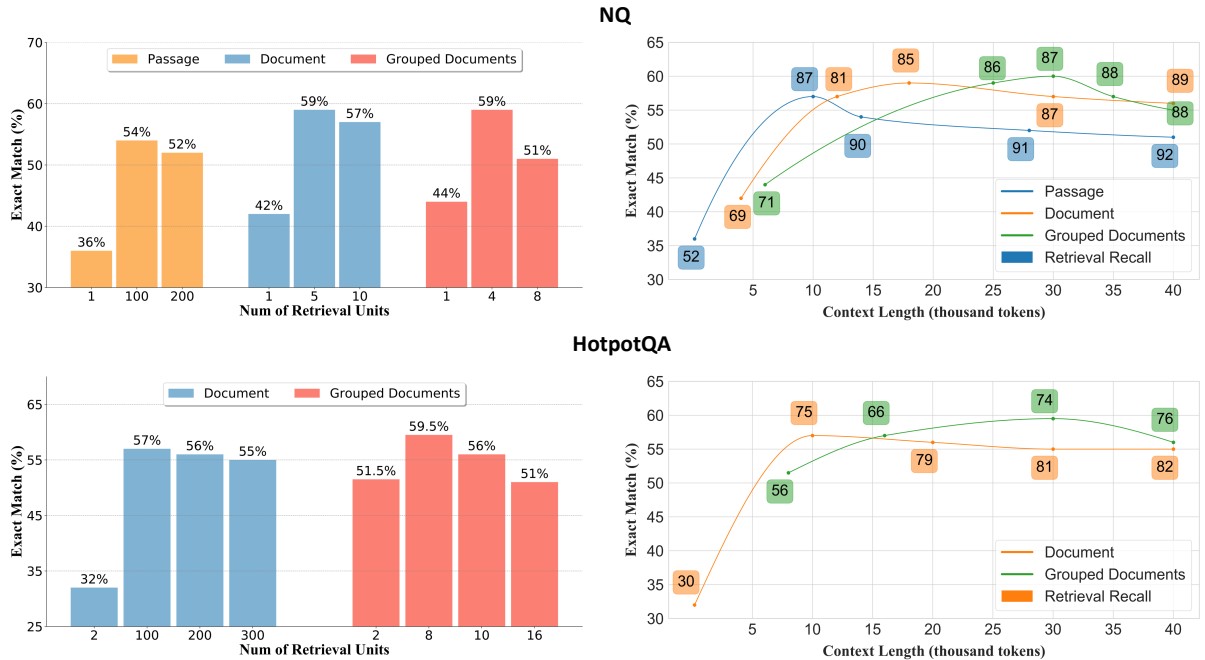

Figure 3: This figure compares different settings of LongRAG, using 200 test cases from the test set to evaluate various retrieval unit selections, demonstrating the effectiveness of our LongRAG design. The upper part of the figure shows the NQ dataset, while the lower part displays the HotpotQA dataset. On the left, it illustrates how the overall performance changes with different settings of retrieval unit size and the number of units fed into the reader; on the right, it shows that the end performance does not increase monotonically with the recall score, and LongRAG is more robust to the influence of "hard negatives" as the context length of the reader increases.

**Retrieval Unit Selection**  Figure 3 compare different retrieval unit settings of LongRAG, specifically focusing on the selection of retrieval unit granularity and the optimal number of retrieval units used in the reader. We have two observations: First, regardless of which retrieval unit is selected, there will be a

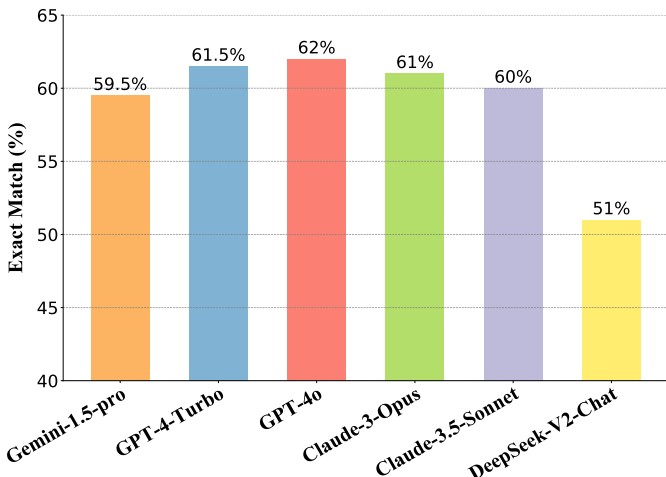

Figure 4: This figure compares different readers of LongRAG on the NQ dataset. This table leverages 200 test cases from the test set to help compare performance using different readers.

turning point where feeding more retrieval units into the reader becomes detrimental. This is due to the excessive burden placed on the reader, preventing it from effectively understanding and extracting relevant information from the long context. Taking NQ as an example: for passage-level retrieval units, the turning point occurs between 100 and 200; for document-level retrieval units, the turning point is between 5 and 10; and for grouped documents level, the turning point is between 4 and 8. In general, the most suitable context length fed into the reader is around 30K tokens. Second, using long retrieval units shows improved performance when comparing passage-level retrieval units with document-level or grouped document-level retrieval units.

**Recall vs. EM**   In Figure 3, we compare the relationship between retrieval recall and end performance across varying context lengths for different retrieval unit selections. We observe that using fewer retrieval units in the reader with longer retrieval units design reduces the introduction of distractors or hard negatives under a given length budget. Consequently, the end performance does not increase monotonically with the recall score. In the future, with advancements in long embedding models and improved retrieval recall for long retrieval units, we can expect better end performance.

**Reader Model**   In Figure 4, we compare the performance of six different readers: Gemini-1.5-pro, GPT-4-Turbo, GPT-4o, Claude-3-Opus, Claude-3.5-Sonnet and DeepSeek-V2-Chat. The results indicate that GPT-4o achieves the highest exact match score on the 200 test questions of the NQ dataset among the three models. This suggests that GPT-4o is the most effective in the role of a long reader in the LongRAG framework. The enhanced performance of GPT-4o can be attributed to its superior ability to process and comprehend lengthy contexts, ensuring that crucial information is accurately extracted. Therefore, we mainly report the GPT-4o results in our main table. Besides, Gemini-1.5-pro, GPT-4-Turbo, Claude-3-Opus, and Claude-3.5-Sonnet could achieve very similar results. These state-of-the-art black box LLMs are also effective readers within the LongRAG framework. Deepseek-V2-Chat is one of the best open-source LLMs, but its performance degrades significantly compared to the previous five black-box LLMs. The above experiments demonstrate that our current framework depends on the long-context understanding ability of LLMs, and we still have a long way to go in harnessing open-source LLMs within our framework.

## 4   Related Work

### 4.1   Retrieval-Augmented Generation.

Augmenting language models with information retrieved from large corpora has become a popular and effective approach for knowledge-intensive tasks, particularly open-domain question answering. The predominant

architecture follows a retriever-reader style (Chen et al., 2017; Guu et al., 2020), where the input query retrieves information from a corpus, and a language model uses this information as additional context to make a final prediction. Recent work has focused on improving the retriever (Karpukhin et al., 2020; Xiong et al., 2020a; Qu et al., 2020; Xiong et al., 2020b; Khalifa et al., 2023), enhancing the reader (Izacard & Grave, 2020b; Cheng et al., 2021; Yu et al., 2021; Borgeaud et al., 2022), fine-tuning the retriever and reader jointly (Yu, 2022; Izacard et al., 2022; Singh et al., 2021; Izacard & Grave, 2020a), and integrating the retriever with the black-box language model (Yu et al., 2023; Shi et al., 2023; Trivedi et al., 2022). However, the impact of document granularity on the effectiveness and efficiency of the retrieval-augmented generation pipeline remains underexplored.

### 4.2 Long Context Large Language Models.

The effectiveness of Transformer-based models is hindered by the quadratic increase in computational cost relative to sequence length, especially when dealing with long context inputs. In order to solve this issue, different approaches have been proposed to mitigate computational issues, including sliding memory window and chunk segmentation (Hao et al., 2022; Ratner et al., 2023; Zhu et al., 2024b). FlashAttention (Dao et al., 2022) has also been a pivotal strategy to significantly reduce the memory footprint to almost linear w.r.t sequence length.

To enable length extrapolation, RoPE (Su et al., 2021) and AliBI (Press et al., 2021) position encodings have shown potential to enable length extrapolation, which have been widely used in the literature. Recent endeavors have explored diverse strategies to tackle this challenge, which is mainly *Position reorganization* (Jin et al., 2024; An et al., 2024), *Position interpolation* (Chen et al., 2023a; Peng et al., 2023; Liu et al., 2024). Furthermore, alternative architectures beyond the Transformer have been explored to handle long inputs more naturally. These diverse approaches claim that they can enhance the capabilities of LLMs in processing long context inputs more efficiently.

### 4.3 Long Context Embedding

Recent efforts also increased the context length for embedding models, extending the supported text snippet length from a limit of 512 tokens to 32k tokens. Typically, the development of long-context embedding models involves first obtaining a long-context backbone model. This can be achieved either by pre-training with long inputs from scratch (Günther et al., 2023; Nussbaum et al., 2024; Chen et al., 2024) or by utilizing existing large language models that support longer context (Wang et al., 2023). Additionally, some works extend the capabilities of existing embedding models to handle long contexts by applying LLM content window extension methods on embedding models (Zhu et al., 2024a; Peng & Quesnelle, 2023), or by employing state-space encoder models (Saad-Falcon et al., 2024).

## 5 Conclusion

In this paper, we propose a new framework, LongRAG, to alleviate the imbalance between the burden of the retriever. The LongRAG framework consists of a "long retriever" and a "long reader" component on top of the 4K-token retrieval units. Our proposed framework can significantly reduce the corpus size, enabling strong retrieval recall using only a few top units, thereby minimizing noise from hard negatives. On the other hand, the long retrieval unit preserves the semantic integrity of each document. We test our framework on four end-to-end question answering tasks and demonstrate its superior performance without any training. We believe LongRAG can pave the road for the modern RAG system design.

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

## A  Appendix

### A.1  Prompts Template for Long Context Reader

We have put out prompts used for the experiments in Table 7. For the closed-book method, we use 16-shot in-context examples. For LongRAG, we use a two-turn approach to extract the final answer. In the first turn, the long retrieved context and the question are concatenated as input, and we do not use any in-context examples here due to the context being around 30K tokens. Empirically, we found it beneficial to let the reader generate a longer answer initially, typically ranging from a few words to a few sentences. In the second turn, we use 8-shot in-context examples to guide the reader in further extracting the most important part of the long answer as the short answer, which is typically just a few words.

| Method | Prompt |
|---|---|
| CLOSED-BOOK | Here are some examples of questions and their corresponding answer, each with a "Question" field and an "Answer" field. Answer the question directly and don't output other thing.
"Question": . . . "Answer": . . .
"Question": . . . "Answer": . . .
. . .
"Question": . . . "Answer": . . .
Answer the following question.
"Question": who is the owner of reading football club "Answer": |
| LONGRAG | **Turn 1:** Go through the following context and then answer the question. The context is a list of Wikipedia documents, ordered by title: . . . .
Each Wikipedia document contains a title field and a text field. The context is:
"Title": . . . "Text": . . .
"Title": . . . "Text": . . .
. . .
"Title": . . . "Text": . . .
Find the useful documents from the context, then answer the question: when did the philadelphia eagles play in the super bowl last. Answer the question directly. Your response should be very concise.
**Turn 2**: You have been provided with a question and its long answer. Your task is to derive a very concise short answer from the given long answer. It's important to ensure that the output short answer remains as simple as possible. Here a few examples:
"Question": . . . "Long Answer": . . . "Short Answer": . . .
"Question": . . . "Long Answer": . . . "Short Answer": . . .
. . .
"Question": . . . "Long Answer": . . . "Short Answer": . . .
Extract the short answer of the following question and long answer:
"Question": when did the philadelphia eagles play in the super bowl last "Long Answer": The Philadelphia Eagles last played in the Super Bowl on February 4, 2018, in Super Bowl LII. "Short Answer": |

Table 7: Here are the prompts we used for all the experiments. For the closed-book method, we use 16-shot in-context examples. For LongRAG, we use a two-turn approach to extract the final answer. The first turn doesn't require any in-context examples and generate a longer answer, typically ranging from a few words to a few sentences. In the second turn, we use 8-shot in-context examples to calibrate and extract the exact short answer, which is typically just a few words.

## A.2 Refined Metric

The most standard metric used in open-domain extractive question answering tasks is EM (Exact Match), since the correct answer must be a substring within the corpus. In our framework, since the long retrieved context, which contains multiple highly-related documents to the given query, is fed into the reader, there is a much higher possibility that an alias of the ground truth exists in the context and can be extracted by the reader. As shown in Table 8, although LongRAG's prediction doesn't exactly match the ground truth, it's obvious that LongRAG's prediction is correct. To better and more fairly evaluate LongRAG's performance, we have refined the EM metric slightly. We recognize it as an exact match if the prediction is less than five tokens (indicating that the short answer is successfully extracted as described in Section A.1) and the ground truth is a substring of the prediction or vice versa. We have also manually verified that this refined metric indeed captures aliases or other forms of the ground truth. For the fully-supervised RAG baselines used in our paper, given that they are fine-tuned on the training data and the retrieval unit is a small snippet, we believe that the difference won't be significant when using the refined EM.

| Question | Ground truth | LongRAG prediction |
|---|---|---|
| where does the bob and tom show broadcast from | Indianapolis , Indiana | Indianapolis |
| who has given the theory of unbalanced economic growth | Hirschman | Albert O. Hirschman |
| when does season 6 of the next step start | 2018 | September 29, 2018 |
| what was the precursor to the present day internet | the ARPANET project | ARPANET |

Table 8: Some examples demonstrate that LongRAG has extracted aliases or different forms of the ground truth.

### A.3 Group Documents Algorithm

In this section, we provide an example algorithm used to formulate long retrieval units by grouping multiple short documents, which we applied in the NQ and HotpotQA experiments in our paper. In the algorithm, whether two documents are related can be determined by any reasonable function, such as hyperlinks, word frequency, or structural information from the dataset. In the two Wikipedia-related question-answering tasks in our paper, NQ and HotpotQA, we use the hyperlinks embedded in the text to describe the relationships between documents.

---

**Algorithm 1** Example Group Documents Algorithm

---

**Input:** $S$ (max number of tokens per group), $D$ (list of documents), adj[$d$] (related documents for each document $d$), deg($d$) (number of related documents for each document $d$)
**Output:** $\mathcal{G}$ (set of groups)
Sort $D$ from low degree to high degree based on deg($d$)
Initialize an empty set of groups $\mathcal{G}$
**for** each document $d$ in $D$ **do**
    related_groups $\leftarrow \emptyset$
    **for** each related document $r$ in adj[$d$] **do**
        **for** each group $g$ in $\mathcal{G}$ **do**
            **if** $r \in g$ **then**
                related_groups $\leftarrow$ related_groups $\cup \{g\}$
            **end if**
        **end for**
    **end for**
    Create a new group $g_{\text{new}} = \{d\}$
    Sort related_groups by their size
    **for** each group $g$ in related_groups **do**
        **if** $|g_{\text{new}}| + |g| \leq S$ **then**
            $g_{\text{new}} \leftarrow g_{\text{new}} \cup g$
            Remove $g$ from $\mathcal{G}$
        **end if**
    **end for**
    Add $g_{\text{new}}$ to $\mathcal{G}$
**end for**
**return** $\mathcal{G}$

---

## A.4  Dataset Examples

Here, we present a few examples from the four datasets we experiment with.

| Method | Prompt |
|---|---|
| NQ | **Question:** how many episodes are in series 7 game of thrones 
 **Answer:** seven |
| HOTPOTQA | **Question:** What government position was held by the woman who portrayed Corliss Archer in the film Kiss and Tell? 
 **Answer:** Chief of Protocol |
| QASPER | **Question:** In the paper 'End-to-End Trainable Non-Collaborative Dialog System', How is intent annotated? 
 **Answer:** using a role-playing task on the Amazon Mechanical Turk platform and collecting typed conversations |
| MULTIFIELDQA-EN | **Question:** What is the name of the most active fan club? 
 **Answer:** South West Ultras fan club. |

Table 9: Here are some examples from the four datasets used in our experiments.

