# OpenReview forum: "LongRAG: Enhancing Retrieval-Augmented Generation with Long-context LLMs"
_TMLR — Rejected by TMLR_

### Review · Reviewer_VXyW · 2024-09-24

**Summary Of Contributions:**

In this paper, the LongRAG framework is proposed to solve the problem of normally short retrieval units in traditional RAG framework. It is mentioned that putting short retrieval units into an LLM will result in the loss of contextual information, which will hurt the overall performance. Therefore, the Long Retrieval Unit is created, and the Long Retriever and Long Reader are also constructed to work with it. Long Retrieval Unit is created by using entire documents or grouping related short documents. In Long Retriever, compared with that of traditional RAG framework, less retrieval units can be recalled and no re-rank module is used. In Long Reader, existing long-context LMs, like Gemini are prompted in a zero-shot fashion. In the experiment, LongRAG, without fine-tuning, achieved a similar effect to fully-trained SOTA model. LongRAG is indeed attempt for combing RAG with long-context LLMs.

**Audience:**

Yes

**Broader Impact Concerns:**

No concerns on the ethical implications of the work.

**Claims And Evidence:**

Yes

**Requested Changes:**

In this paper, the Long RAG framework is propose. The ideas and experiments are innovative. However, the weaknesses are requested to fix.

1. Table 4 compares the performance of different encoder models at different granularities, but GBE-Large does not provide results at large granularities, while E5-Mistral-7B does not provide results at small granularities. (critical)

2. The experiments and discussion of the re-rank module are necessary to be supplemented. (simply strengthen)

3. Work on &quot;without fine-tuning&quot; is rarely mentioned in related workers and experiments. (simply strengthen)

4. In view of the key problem raised in this paper, &quot;short retrieval unit will cause LLM to lose the original context information&quot;, there are also few investigations and experiments on the solution and it is highly recommended to supplement them. (critical)

5. In the discussion of Similarity Search in section 2.1, the similarity calculation method is very similar to that of small-to-big(Wang et al., 2024), and it is recommended to add an explanation of the difference between the two to avoid misunderstanding. (critical)

Reference:

Xiaohua Wang, Zhenghua Wang, Xuan Gao, Feiran Zhang, Yixin Wu, Zhibo Xu, Tianyuan Shi, Zhengyuan Wang, Shizheng Li, Qi Qian, Ruicheng Yin, Changze Lv, Xiaoqing Zheng, Xuanjing Huang. Searching for Best Practices in Retrieval-Augmented Generation. *arXiv preprint arXiv:*2407.01219. 2024

**Strengths And Weaknesses:**

Strengths
In this paper, the problem that short retrieval units will cause LLMs to lose the original context information during inference is propose, and it can be solved by increasing the length of the retrieval units. In order to accommodate long retrieval units, the Long Retriever and Long Reader are constructed. Experiments show that the performance of LongRAG is close to fully-trained SOTA model. This idea has been fully verified in the context of in-depth research on long-text embedding and long-text LLMs, which is worthy recognition.

In addition, Grouping short documents is a smart solution to formulate long retrieval units. It can not only effective increase the length of the documents and the coherence of the context, but also may promote the repeated verification of knowledge by large models and import the reliability of reasoning. Referring to Self-Consistency(Wang et al., 2023), the input of multiple similar knowledge strengthens the validation of knowledge by LLMs.

The ablation studies has references value, which fully compares the effects of the size of retrieval units and the length of the context on the effect in LongRAG.

Weaknesses
There are also some problems, which should be solved before it is considered for publication. If the following problems are well-addressed, the reviewer believes that the essential contribution of this paper are important for long-context RAG research.

The experiment still needs to be improved. In Table 4, the AR@1 of different encoder models at different granularities are compared. However, BGE-Large does not provide result at large granularities, while E5-Mistral-7B does not provide results at small granularities. Moreover, experiments at smaller granularities than 512 tokens chunks are also worth doing. On the other hand, as Mentioned in Introduction, re-rank is not used but it has not been fully experimented. Re-rank can not only be used to filter results to improve search accuracy, but also help large models better understand knowledge and improve the quality of responses(Liu et al., 2024).

The review of related work needs to be supplemented. Long RAG "without fine-tuning" is a key advantage in this paper but it is not fully discussed in related work. Additionally, the problem raised in this paper is that short retrieval units will cause LLMs to lose the context information when reasoning, and there are some other solutions for this problem such as Adaptive-RAG (Jeong et al., 2024). Adaptive-RAG continuously improves the answer by allowing the LLMs and the retriever to interact multiple times.

In section 2.1, in the chapter on Similarity Search, it is mentioned that the maximum value of similarity under multiple granularity chunks will be used as the similarity between the retrieval unit and query. There seems to be similar to the idea of small-to-big(Wang et al., 2024) and this can be considered to quote or elaborate on the difference.

Reference
Nelson F. Liu, Kevin Lin, John Hewitt, Ashwin Paranjape, Michele Bevilacqua, Fabio Petroni, Percy Liang. Lost in the Middle: How Language Models Use Long Contexts. Trans. Assoc. Comput. Linguistics 12: 157-173, 2024.

Soyeong Jeong, Jinheon Baek, Sukmin Cho, Sung Ju Hwang, Jong Park. Adaptive-RAG: Learning to Adapt Retrieval-Augmented Large Language Models through Question Complexity. In NAACL-HLT* 2024 the North American Chapter of the association for Computational
Linguistics: Human Language Technologies*: 7036-7050, 2024.

Xiaohua Wang, Zhenghua Wang, Xuan Gao, Feiran Zhang, Yixin Wu, Zhibo Xu, Tianyuan Shi, Zhengyuan Wang, Shizheng Li, Qi Qian, Ruicheng Yin, Changze Lv, Xiaoqing Zheng, Xuanjing Huang. Searching for Best Practices in Retrieval-Augmented Generation. *arXiv preprint arXiv:*2407.01219. 2024

Xuezhi Wang, Jason Wei, Dale Schuurmans, Quoc V. Le, Ed H. Chi, Sharan Narang, Aakanksha Chowdhery, Denny Zhou. Self-Consistency Improves Chain of Thought Reasoning in Language Models. In ICLR *2023 International Conference on Learning Representations*, 2023.

---

### Review · Reviewer_mRWC · 2024-09-24

**Summary Of Contributions:**

The paper introduces LongRAG, a RAG framework that addresses the imbalance between retriever and reader in traditional RAG systems. Instead of using short 100-word units for retrieval, LongRAG processes longer 4K-token units by grouping related documents, reducing the total number of units and the burden on the retriever. LongRAG shows promise for combining long-context LLMs with retrieval-augmented systems, improving accuracy on several benchmarks, including NQ, HotpotQA, Qasper and MultiFieldQA-en.

**Audience:**

Yes

**Claims And Evidence:**

Yes

**Requested Changes:**

See weaknesses

And a few typos:
- In Introduction，“This design choice was made in an era when the reader models were heavily restricted by their ability to **handle long and contexts.**”
- In 2.1，"In traditional RAG, CF is usually small which contains about **hundred** of tokens, which should"
- In Figure 2 caption，"... depending on the **orinal** docuemnt size. In ..."

**Strengths And Weaknesses:**

Strengths:
 - The method is quite simple, easy to understand and training free.

Weaknesses:
- I am concerned about whether the performance improvements are due to unifying many short units into one long unit, or simply because the large language models (i.e., Gemini, GPT-4)  used as readers in this study are inherently more powerful. It appears that the baseline reader model in this paper is not one of the state-of-the-art LLMs. To ensure a fair comparison and validate the effectiveness of the proposed method, I suggest replacing the baseline reader with the same LLMs used in this work.
- When grouping related documents, the authors seem to only use the hyperlinks embedded in the text to describe relationships between documents. The function determining whether two documents are related is important. If key short retrieval units containing the answer are assigned to different long retrieval units, this could lead to the retrieved top long units missing the critical information needed to answer the query, which may ultimately hurt overall performance. I suggest that experimenting with different document grouping functions to assess the impact on the final results and verify the robustness of the proposed method.

---

### Review · Reviewer_w4eh · 2024-10-25

**Summary Of Contributions:**

The paper introduces LongRAG, a novel framework that addresses the imbalance between retrievers and readers in traditional RAG systems by utilizing longer retrieval units (4K tokens) instead of short passages (100-200 tokens). The authors propose three key components: long retrieval units formed by grouping related documents or using whole documents, a simplified long retriever that identifies relevant information by searching through these units, and a long reader that processes the concatenated retrievals (≈30K tokens) using modern LLMs. Through extensive experimentation on four datasets (NQ, HotpotQA, Qasper, and MultiFieldQA-en), they demonstrate that LongRAG achieves competitive performance without any training, matching or approaching state-of-the-art supervised models. For Wikipedia-based datasets, they reduce the corpus size from 22M to 600K units while improving retrieval performance, achieving 62.7% EM on NQ and 64.3% on HotpotQA. For non-Wikipedia datasets with naturally longer documents, processing entire documents as single units improves F1 scores by 3.4% on Qasper and 6.3% on MultiFieldQA-en compared to traditional chunking approaches.

**Audience:**

Yes

**Claims And Evidence:**

No

**Requested Changes:**

Critical Changes (Required for Acceptance):

1. Memory and Computational Analysis
- Add a comprehensive analysis of memory requirements and computational costs comparing traditional RAG vs LongRAG
- Include benchmarks showing index sizes, retrieval latencies, and memory footprint during inference
- Provide concrete throughput numbers (queries/second) for different retrieval unit sizes

This is critical because deployment feasibility is a key consideration for the proposed method.

2. Position Bias Investigation
- Add experimental analysis showing answer extraction accuracy as a function of position in concatenated retrievals
- Compare performance when relevant information appears in different positions within the 30K token window
- Include position-aware evaluation metrics

This is necessary to validate the reliability of the long reader component.

3. Generalized Document Grouping
- Develop and evaluate alternative document relationship metrics beyond hyperlinks
- Include evaluations on at least one additional domain without explicit link structure
- Provide concrete guidelines for relationship threshold selection

This is critical for demonstrating broad applicability.

Important but Non-Critical Changes:

4. Query-Document Length Analysis
- Add experiments varying query length and studying its impact on retrieval quality
- Consider query expansion techniques to bridge the length gap
- Compare performance with and without query preprocessing

5. Similarity Computation Enhancement
- Experiment with alternatives to MaxP for long document similarity
- Consider hierarchical or multi-scale similarity computation approaches
- Analyze cases where cross-chunk relationships are important

6. Ablation Studies
- Include more granular analysis of retrieval unit size impact (e.g., 1K, 2K, 4K, 8K tokens)
- Study the effect of different document grouping criteria
- Evaluate different concatenation strategies for retrieved documents

7. Scalability Analysis
- Test performance on larger corpora (>10M documents)
- Analyze index update costs with long retrieval units
- Include distributed retrieval experiments

**Strengths And Weaknesses:**

Strengths :-
- The approach effectively reduces index size and retrieval complexity while maintaining or improving performance, demonstrating clear practical benefits.
- The extensive ablation studies provide strong empirical evidence for the advantages of longer retrieval units, especially in reducing hard negatives.
- The framework's zero-shot nature and ability to work with off-the-shelf components makes it highly practical and easily adoptable.
- The evaluation across diverse datasets (both Wikipedia and non-Wikipedia) demonstrates broad applicability.
- The technical implementation details, especially the MaxP design for long context similarity computation, are well thought out and practical.

Weaknesses :-

- While the paper discusses reducing corpus size, it doesn't adequately address the memory implications of storing and processing 4K-token embeddings versus traditional 100-200 token embeddings. The MaxP approximation for similarity computation suggests potential memory bottlenecks that aren't thoroughly analyzed.

- The document grouping algorithm (Appendix A.3) relies heavily on hyperlinks for Wikipedia data, but this approach may not generalize well to other domains where such explicit relationships aren't available. The paper doesn't provide robust alternatives for relationship determination in general cases.

- There's an inherent asymmetry between short queries (typically 10-20 tokens) and long document units (4K tokens) that isn't addressed. This mismatch could affect semantic matching quality, especially given that modern embedding models are typically trained on more balanced length pairs.

- The strong results rely heavily on proprietary black-box LLMs (GPT-4, Gemini 1.5). The significant performance drop with open-source models (e.g., DeepSeek-V2-Chat showing ~10% lower performance) raises concerns about the framework's broader accessibility and reproducibility.

- The paper doesn't address potential position bias issues in the long reader when processing 30K tokens of concatenated retrievals. Given that even modern LLMs can exhibit position bias with long inputs, this could affect answer extraction reliability, especially when relevant information appears later in the concatenated sequence.

- The paper's approach of using max pooling over chunk similarities (MaxP design) might miss important cross-chunk semantic relationships, potentially leading to suboptimal retrieval in cases where the answer requires understanding relationships across different parts of a long document.

---

### Decision · Action_Editor_5fpz · 2024-12-14

**Recommendation:** Reject

**Comment:**

This paper introduces the LongRAG framework to solve the problem of normally short retrieval units in traditional RAG framework. It received scores of Leaning Accept, Leaning Reject, and Reject recommendations. On the one hand, reviewers commented that (1) the method is simple, easy to understand and training-free, and (2) the idea has been well verified in the experiments. On the other hands, all the 3 reviewers have provided comments where the experiments and analysis can be further improved to make the paper more convincing, e.g., the performance improvements are due to unifying many short units into one long unit, or simply due to stronger LLMs being used. However, due to the leading author's personal reasons, no rebuttal is provided, leaving all the concerns and requested changes unaddressed. Therefore, the Action Editor decided to recommend rejection by the end.

**Audience:**

Yes.

**Claims And Evidence:**

Yes, but the experiments and analysis can be further enhanced to make the claims more convincing.

**Resubmission Of Major Revision:**

The authors may consider submitting a major revision at a later time.